# High-Frequency Conductivity of Amorphous and Crystalline Sb$_2$Te$_3$ Thin Films

Rene Castro *, Aleksei Kononov and Nadezhda Anisimova

Herzen State Pedagogical University of Russia, Institute of Physics, 48 Moika Emb., 191186 St. Petersburg, Russia; kononov_aa@icloud.com (A.K.)
* Correspondence: recastro@mail.ru

**Abstract:** The results of study of charge transfer processes in thin amorphous and crystalline Sb$_2$Te$_3$ films in a wide range of frequencies and temperatures are presented. The frequency spectra of conductivity were obtained by the dielectric spectroscopy method. The authors analyzed the frequency dependences of the conductivity in the electric field and the temperature dependences of the exponent $s$. A transition from the classical correlated barrier hopping (CBH) to quantum mechanical tunneling (QMT) was observed at a certain temperature $T_t$. The CBH model allowed the authors to calculate the conductivity parameters of two phases. Two areas with different types of conductivity were revealed on the conduction spectra, and the activation energies of charge transfer processes for amorphous and crystalline films were determined. The following features were discovered: the difference in the temperatures of the change of the charge transfer mechanism and the transition from the semiconductor region to the metal region on the temperature dependence of conductivity. They can help to identify the amorphous phase in the quasi-binary chalcogenide Sb$_2$Te$_3$-GeTe system.

**Keywords:** functional chalcogenides; Sb$_2$Te$_3$; phase transition; crystalline and amorphous films

## 1. Introduction

Chalcogenide glassy semiconductors (CGS) attract the attention of researchers due to their use in numerous components and sensors in micro- and optoelectronics. For example, in modern times, CGS are used in the manufacture of thermal imaging systems [1], fibers and planar waveguides that are transparent in the IR range [2], in optical sensors [3] and nonlinear optics [4]. They are considered to be promising for creating solar cell elements [5].

Binary compounds, such as A$_2^{V}$B$_3^{VI}$, were already well studied by various methods, and antimony telluride is a typical representative of them [6,7]. Nowadays, they are of great practical interest [8–13]. Sb$_2$Te$_3$ is a narrow-gap semiconductor with a rhombohedral structure. The features of this structure and the presence of alternating layers lead to the anisotropy of many properties. The authors [14,15] studied the dielectric properties of amorphous and polycrystalline Sb$_2$Te$_3$ of various thicknesses ($d = 20 \ldots 600$ nm) obtained by thermal evaporation in vacuum. The observed temperature dependences of the conductivity and the exponent of its frequency dependence $s$ were interpreted within the framework of the classical correlated barrier hopping model (CBH model) [16]. The authors [16] found a trend towards an increase in the amorphous fraction in films with an increase in their thickness. However, in all the above-mentioned works, there was no interpretation of the experimental results, leading to any connection between the detected conductive properties and the structure of the material.

The purpose of this work was to reveal the features of charge transfer processes in amorphous and crystalline layers of the Sb$_2$Te$_3$ chalcogenide system obtained by high-frequency magnetron sputtering, and to calculate the conductivity parameters of the systems under study. The study aimed to determine how a change in the dimensionality of the structure (passing from three-dimensional amorphous to two-dimensional crystalline) affects the process of high-frequency conduction.

## 2. Theory

The total conductivity $\sigma_{tot}(\omega)$ measured in a given experiment at particular temperature can be written as:

$$\sigma_{tot}(\omega) = \sigma(\omega) + \sigma_{dc} \tag{1}$$

where $\sigma_{dc;}$ and $\sigma(\omega)$ are the direct current (dc) and frequency-dependent (ac) conductivities, respectively.

In many amorphous semiconductors and insulators, the ac conductivity invariably has the form:

$$\sigma(\omega) = A(T){\cdot}\omega^s, \tag{2}$$

where $A$ is a constant dependent on temperature and the exponent $s$ is generally less than or equal to unity ($0.5 < s < 1.0$).

Many different theories were proposed for ac conduction in amorphous and crystalline semiconductors. It is usually assumed that the pair approximation is preserved, i.e., the fact that the occurrence of dielectric losses is due to the movement of a charge carrier between two equilibrium positions is accepted [16,17]. Two conduction mechanisms are the most useful: quantum mechanical tunneling and the classical jump over the barrier, but some combination of these two options is also possible. According to the CBH model [16], electrons jump between energy states, overcoming the potential barrier. In this case, the expression for ac conductivity for a specific fixed temperature has the form [18]:

$$\sigma(\omega) = \frac{\pi^3 N^2 \varepsilon \varepsilon_0 \omega R_\omega^6}{24}, \tag{3}$$

where $N$ is the density of states between which charge carriers jump. The relationship between the jump length $R_\omega$ and the height of the potential barrier is expressed by the relationship:

$$R_\omega = \frac{e^2}{\pi \varepsilon \varepsilon_0} \left[ W_M - kT \ln\left(\frac{1}{\omega \tau_0}\right) \right]^{-1} \tag{4}$$

where $W_M$ is the barrier height, $\tau_0$ is the characteristic relaxation time, the reciprocal of the phonon frequency $\nu_{ph}$. On the other hand, the exponent $s$ is related to the barrier height $W_M$ via the expression:

$$s = 1 - \frac{6kT}{W_M}. \tag{5}$$

Austin and Mott proposed a model according to which charge transfer occurs via thermally activated quantum mechanical tunneling of charge carriers (QMT model) [19]. The frequency dependence of conductivity obeys the law (3), and the expression for ac conductivity (2) has the form:

$$\sigma(\omega) = Ce^2 k_B T \alpha^{-1} [N(E_F)]^2 \omega R_\omega^4, \tag{6}$$

where $C$—constant, which is usually taken to be $\pi^4/24$ [20], $N(E_F)$—density of localized states at the Fermi level, $R_\omega$—jump length at a specific frequency $\omega$.

Within the framework of the pairwise approximation, it is assumed that the carriers form non-overlapping small polarons [18], i.e., the total charge carrier energy is reduced by the polaron energy $W_p$ resulting from the lattice distortion. The transfer of an electron between degenerate regions will be associated with the activation energy, that is, the polaron jump energy equal to $W_H \cong W_p/2$. The expression for the exponent $s$, which implies an increase in its value with increasing temperature, will look like:

$$s = 1 - \frac{4}{\ln(1/\omega \tau_0) - W_H/k_B T}. \tag{7}$$

The tunneling distance at the frequency $\omega$ in the model of a small-radius polaron becomes equal to:

$$R_\omega = \frac{1}{2\alpha}[\ln(1/\omega\tau_0) - W_H/k_BT]. \tag{8}$$

## 3. Materials and Methods

Amorphous and crystalline $Sb_2Te_3$ thin films were obtained by RF magnetron sputtering. Amorphous layers of 80 nm thick were obtained by sputtering onto a silicon substrate at room temperature. $Sb_2Te_3$ crystalline films were obtained in a two-stage way: they were first grown and were then heated to 230 °C in a sputtering chamber. The amorphism and crystallinity of the obtained films were confirmed by X-ray diffraction [21].

The frequency dependences of the conductivity of the layers under study at different temperatures were obtained with Novocontrol Technologies "Concept-81" spectrometer (Novocontrol Technologies GmbH & Co. KG, Montabaur, Germany; "Modern physical and chemical methods of formation and study of materials for the needs of industry, science and education", Herzen University). This spectrometer was designed to study the electrophysical properties of a wide class of materials [22–24]. The measurements were carried out in the frequency range $f = 10^{-2} \dots 10^5$ Hz and temperatures $T = 243 \dots 353$ K. The amplitude of the voltage applied to the samples was $U = 0.1$ V. The values of the imaginary and real parts of the impedance of the cell with the measured sample were used as experimental data:

$$Z^*(\omega) = R + \frac{1}{i\omega C} = Z' + iZ'' = \frac{U_0}{I^*(\omega)}. \tag{9}$$

The complex conductivity spectra were calculated from the impedance spectra using the formula:

$$\sigma^* = \sigma' - i\sigma'' = \frac{-iS}{\omega Z^*(\omega)d}. \tag{10}$$

The relative error of the experiment did not exceed 5%.

## 4. Results and Discussion

Figures 1 and 2 show the specific conductivity spectra of the layers under study obtained in the dark measurement mode for various temperatures. As follows from the figures, the dispersion $\sigma'$ obeyed the power law (2), which is characteristic of many chalcogenide glassy and amorphous semiconductors [20]. The temperature dependence of $s$ can be used to reveal the features of charge transfer processes in alternating current.

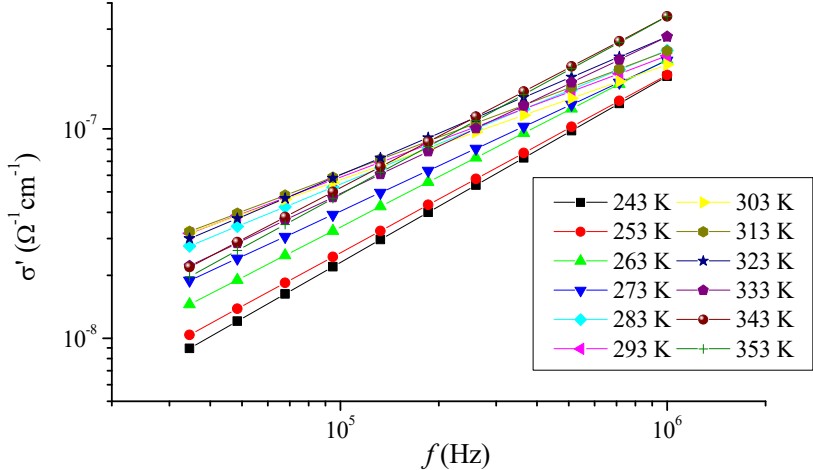

**Figure 1.** Frequency dependence of the specific conductivity $\sigma'$ of amorphous films of the $Sb_2Te_3$ chalcogenide system.

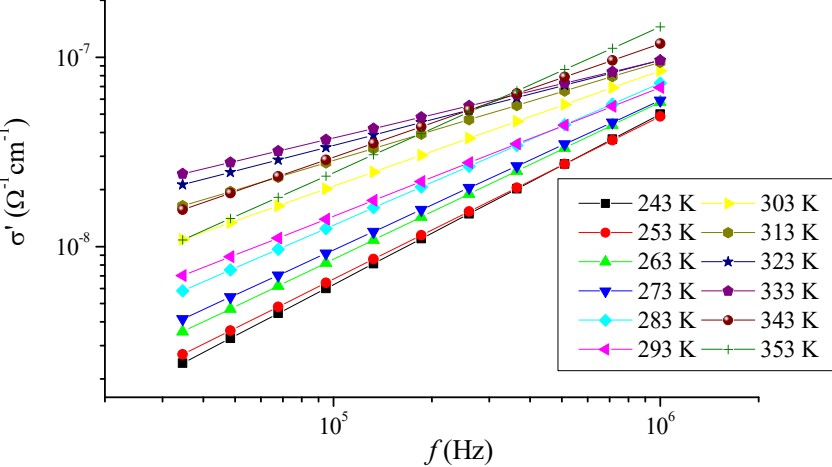

**Figure 2.** Frequency dependence of the specific conductivity $\sigma'$ of crystalline films of the $Sb_2Te_3$ chalcogenide system.

The temperature dependence of the parameter $s$ indicated the predominant transport mechanism (classical or quantum) and allowed calculating the conductivity parameters of the material within the framework of specific models. As can be seen from Figures 3 and 4, both for amorphous samples and for crystalline, $s$ equals 0.40 . . . 0.90. Two temperature sections were observed: 1—decrease in the value of the exponent with increasing temperature and 2—increase in the value of $s$ with increasing temperature.

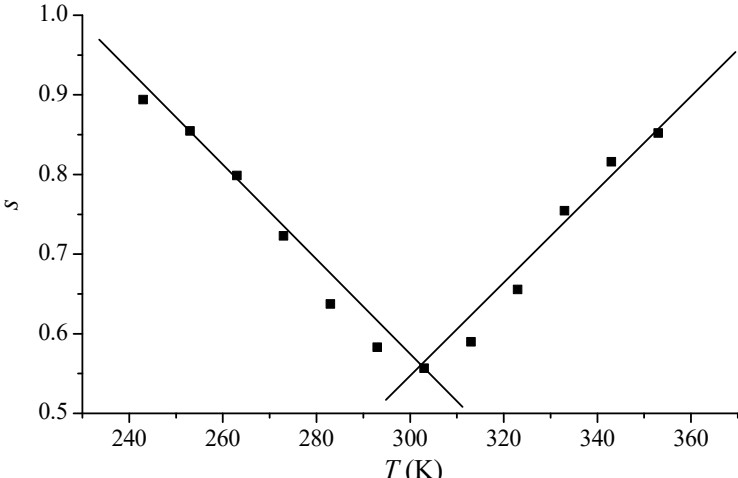

**Figure 3.** Temperature dependence of the exponent $s$ for amorphous layers of chalcogenide $Sb_2Te_3$.

According to modern theoretical concepts, the decrease in $s$ with temperature increase corresponds to the classical jump transition through the barrier described by the CBH model and, accordingly, by Equations (3)–(5) [16]. It was assumed that the charge transfer was carried out by means of electron jumps through the potential barrier $W$ between two localized states (equilibrium centers). In this case, the barrier height was determined by the Coulomb interaction between neighboring defect states. In the case of the systems under study, charged structural defects forming a dipole can act as defect states.

As was shown in our previous study [25], based on the obtained experimental data, it was possible to calculate the values of the conductivity parameters $N$, $R_\omega$ and $W_M$ at different temperatures using Equations (3)–(5). The results of the calculations are presented in Table 1. To determine the value of $R_\omega$, the frequency equal to $3.5 \times 10^4$ Hz was used.

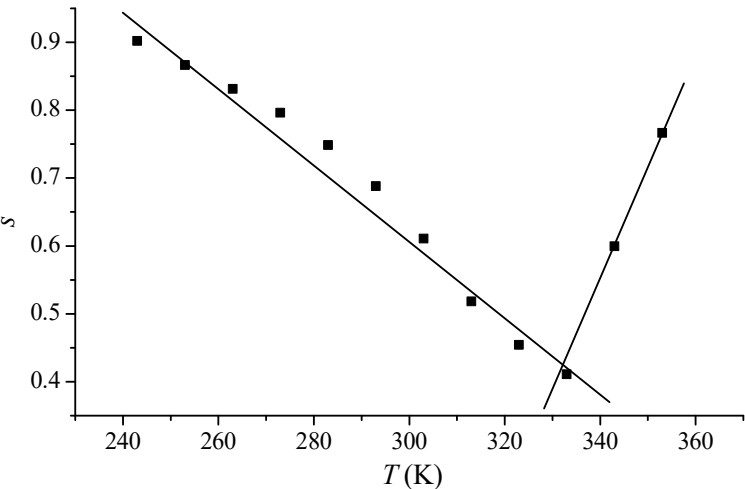

**Figure 4.** Temperature dependence of the exponent *s* for crystalline layers of chalcogenide $Sb_2Te_3$.

**Table 1.** The value of charge transfer parameters in amorphous and crystalline samples of $Sb_2Te_3$.

| *T* (K) | *s* | *N* (m$^{-3}$) | $R_\omega$ (Å) | $W_M$ (eV) |
|---------|-----|----------------|----------------|------------|
| | | amorphous material | | |
| 263 | 0.79 | $2.73 \times 10^{26}$ | 3.26 | 0.68 |
| 273 | 0.72 | $1.31 \times 10^{25}$ | 9.44 | 0.51 |
| 283 | 0.64 | $3.45 \times 10^{23}$ | 33.35 | 0.40 |
| 293 | 0.58 | $1.19 \times 10^{25}$ | 10.57 | 0.36 |
| 303 | 0.56 | $2.85 \times 10^{25}$ | 7.95 | 0.35 |
| | | crystalline material | | |
| 263 | 0.83 | $1.71 \times 10^{26}$ | 3.22 | 0.81 |
| 273 | 0.79 | $6.17 \times 10^{25}$ | 4.69 | 0.69 |
| 283 | 0.75 | $1.16 \times 10^{25}$ | 8.52 | 0.58 |
| 293 | 0.69 | $2.56 \times 10^{23}$ | 31.90 | 0.49 |
| 303 | 0.61 | $9.28 \times 10^{23}$ | 21.97 | 0.40 |

An increase in the value of the exponent *s* with increasing temperature can be explained by thermally activated quantum mechanical tunneling of charge carriers (QMT model) [19]. In this case, the expression for the frequency dependence of the conductivity had the form (6). The change in the charge transfer mechanism from the classical jump transition through the barrier to mechanical tunneling was theoretically predicted by Elliot [26] and was explained by the transition from carrier hopping near the Fermi level to their implementation mainly near the band edges.

Figures 3 and 4 show that the transition from the classical transport mechanism to quantum mechanical tunneling proceeded at a certain temperature $T_t$. For an amorphous sample, $T_t$ = 303 K, while for a crystalline sample, $T_t$ = 333 K.

The temperature dependence of conductivity in the coordinates $\sigma'(\omega) = \zeta \, (10^3/T)$ for two samples is shown on Figures 5 and 6. They demonstrated that charge transfer was a thermally activated process. The existence of two sections was found: (1) an increase in conductivity with increasing temperature (semiconductor section); (2) decrease in conductivity with increasing temperature (section of metallic conductivity). The observed dependence of the semiconductor–metal transition temperature $T_p$ on the frequency of the applied field is shown in Figure 7 for an amorphous sample. It is interesting that the transition temperature $T_p$ for the lowest frequency f = $3.5 \times 10^4$ Hz coincided for two materials with temperature $T_t$.

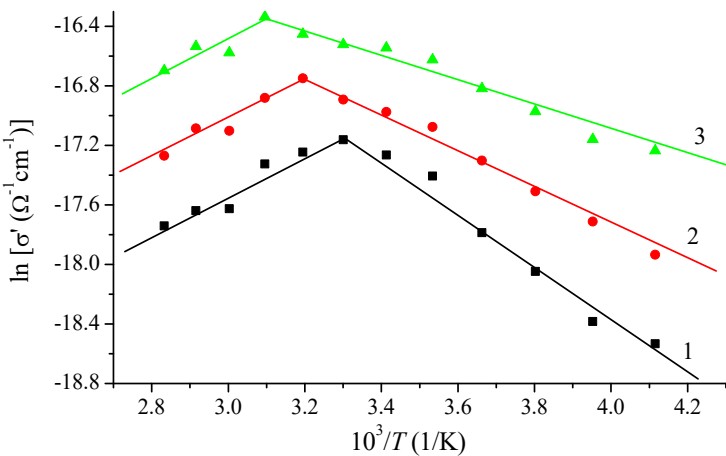

**Figure 5.** Temperature dependence of specific conductivity $\sigma'$ of amorphous $Sb_2Te_3$ films at different frequencies. 1—$f = 3.5 \times 10^4$ Hz, 2—$f = 6.8 \times 10^4$ Hz, 3—$f = 1.3 \times 10^5$ Hz.

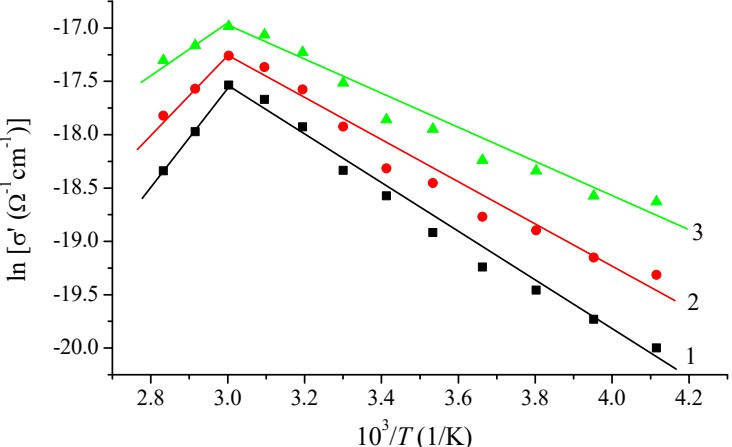

**Figure 6.** Temperature dependence of specific conductivity $\sigma'$ of $Sb_2Te_3$ crystalline films at different frequencies.1—$f = 3.5 \times 10^4$ Hz, 2—$f = 6.8 \times 10^4$ Hz, 3—$f = 1.3 \times 10^5$ Hz.

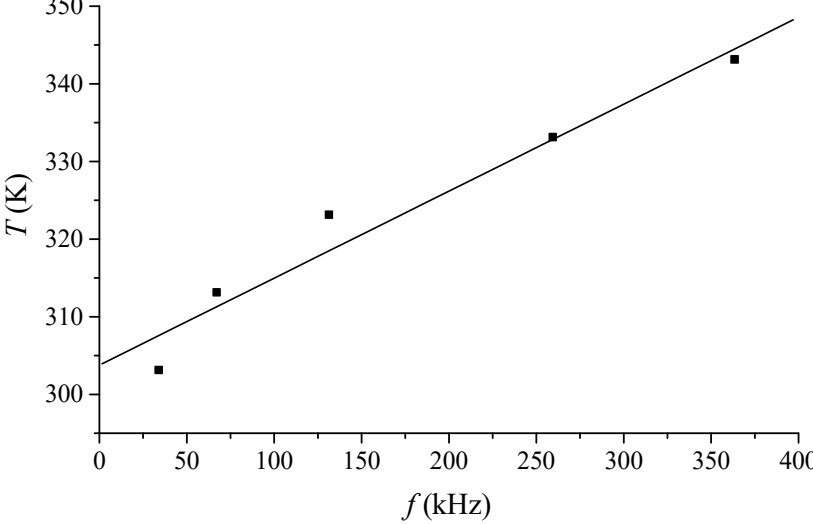

**Figure 7.** Frequency dependence of the semiconductor–metal transition temperature for amorphous $Sb_2Te_3$ films.

For the area of semiconductor conductivity, the activation energy was calculated with the Arrhenius equation:

$$\sigma = \sigma_0 \exp\left(-\frac{\Delta E_\sigma}{kT}\right),\tag{11}$$

where $\sigma_0$—constant, equal to conductivity at direct current.

The frequency dependence of the activation energy $\Delta E_\sigma$ is shown in Figure 8. It was clear that $\Delta E_\sigma$ decreased exponentially with frequency. A similar situation was observed for a wide class of systems, including amorphous structures [27]. Increasing the frequency of the electric field enhanced the effect of electronic transitions between localized states; therefore, the activation energy $\Delta E_\sigma$ decreased with increasing frequency. For the lowest frequency $f = 3.5 \times 10^4$ Hz, the following activation energies were calculated for amorphous and crystalline films, respectively: $\Delta E_\sigma = (0.15 \pm 0.01)$ eV and $\Delta E_\sigma = (0.19 \pm 0.01)$ eV.

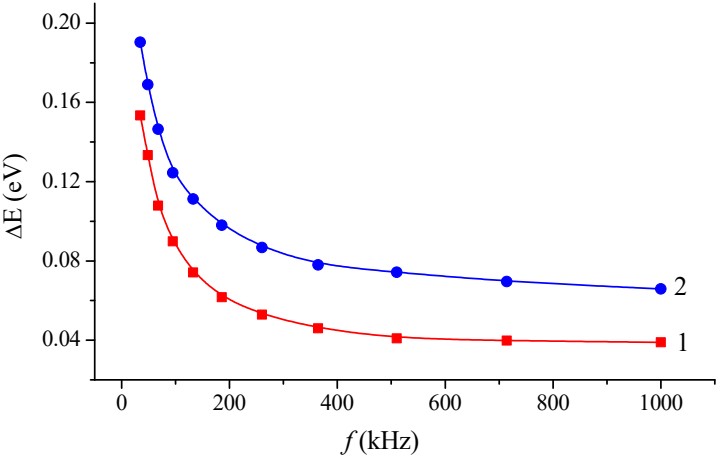

**Figure 8.** Frequency dependence of the activation energy $\Delta E_\sigma$ for Sb$_2$Te3 films. 1—amorphous film, 2—crystalline film.

There is practically no information in the scientific literature about the change in the charge transfer mechanism in thin layers of the Sb$_2$Te$_3$ chalcogenide system. Based on the results of [15], it can be concluded that the change in the activation energy with change in frequency is nonmonotonic, the decrease in the activation energy $\Delta E_\sigma$ at a certain frequency stops and its sharp increase begins. In addition, the existence of two areas was found on the temperature dependence of the density of localized states $N(E)$ for amorphous Sb$_2$Te$_3$ films.

Thus, during the transition from an amorphous structure to a crystalline one, the following changes occurred: a change in the transition temperature from the correlated barrier hopping (CBH) to quantum mechanical tunneling (QMT), a change in the semiconductor–metal transition temperature and an increase in the activation energy.

It is important to consider that, in terms of their nature, the processes in the amorphous and crystalline phases proceed in the same way, despite significant differences in the structure. While the amorphous phase was three-dimensional (3D), the crystal was a layered 2D structure in which covalently bonded blocks were held together by weak Van der Waals bonds. The obtained results suggested that the processes of high-frequency conduction were determined not by the long-range order of the structure, but by its short-range order. In this regard, it is interesting to note that both amorphous and crystalline Sb$_2$Te$_3$ were characterized by the existence of extended ... Te-Sb-Te-Sb ... fragments, with multicenter asymmetric hyperbonds having a dipole moment (Figure 9) [28]. It is noteworthy here that the shorter and longer bonds in such fragments were very similar in the amorphous and crystalline phases. Hence, one should expect general regularities in the behavior of both phases in the presence of an external action (a change in the transfer mechanism, the presence of sections of both semiconductor and metallic conductivity).

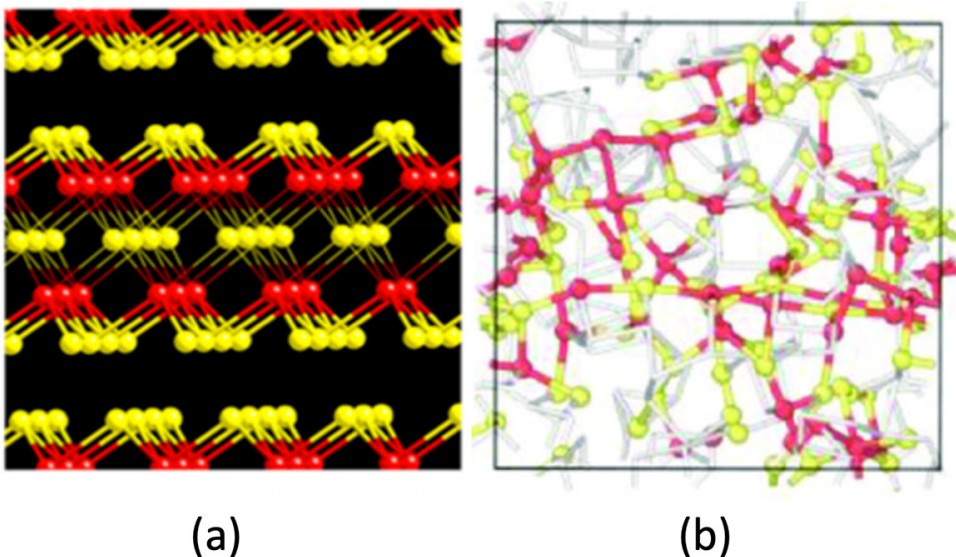

**Figure 9.** Fragments of (**a**) crystalline and (**b**) amorphous structures of $Sb_2Te_3$ [29].

The scholars hold the view that, despite the fact that the stable structure of the crystalline phase of $Sb_2Te_3$ is 2D layered, the existence of a metastable cubic phase with a high concentration of vacancies is also found [30]. It can be assumed that the 3D cubic phase is closer to the 3D amorphous phase. It was noted that the local environments in the 2D-layer phase and in the three-dimensional cubic phase differed from each other, namely in the two-dimensional phase, octahedral fragments with central Sb atoms were surrounded by six Te atoms, while in the three-dimensional phase, there were defective octahedral regions with a smaller number of Te neighbors [30]. A larger number of surrounding Te atoms can be a source of heavier structural units and, accordingly, higher temperatures for the development of polarization processes in the crystalline phase.

### 5. Conclusions

This paper presented the results of a study of charge transfer processes in thin amorphous and crystalline $Sb_2Te_3$ films in a wide range of frequencies and temperatures. A power-law nature of the conductivity dependence on frequency and a decrease in the exponent *s* with increasing temperature were found; these factors indicated the existence of a hopping mechanism of conduction. A transition from the classical correlated barrier-hopping mechanism (CBH) to quantum mechanical tunneling (QMT) at a certain temperature $T_t$ was found, while for amorphous materials, $T_t = 303$ K and for crystalline layers, $T_t = 333$ K. Within the framework of the CBH model, the conductivity parameters of two phases were calculated.

Charge transfer in $Sb_2Te_3$ films is a thermally activated process; the existence of two regions on the temperature dependence of conductivity was found: the region of increase in conductivity with increasing temperature (semiconductor region) and the region of decrease in conductivity with increasing temperature (metallic conductivity region). For the lowest frequency, the values of activation energy were obtained: $\Delta E_\sigma = (0.15 \pm 0.01)$ eV and $\Delta E_\sigma = (0.19 \pm 0.01)$ eV for amorphous and crystalline films, respectively.

**Author Contributions:** Conceptualization, R.C.; methodology, R.C., A.K. and N.A.; analysis, R.C., A.K.; writing—original draft preparation, R.C.; writing—review and editing, A.K. and N.A. All authors have read and agreed to the published version of the manuscript.

**Funding:** This work was supported by the Russian Science Foundation (22-19-00766).

**Data Availability Statement:** The data presented in this study are available in this article.

**Acknowledgments:** The authors are grateful to Y. Saito for his participation in the work at its initial stage, as well as to A.V. Kolobov for discussion of the obtained results.

**Conflicts of Interest:** The authors declare no conflict of interest.

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
