# Peer review of "High-Frequency Conductivity of Amorphous and Crystalline Sb2Te3 Thin Films"

_coatings, doi:10.3390/coatings13050950_

Round 1
Reviewer 1 Report
The authors present their results of a study of charge transfer processes in thin amorphous and crystalline Sb2Te3 films in a wide range of frequencies and temperatures. The frequency spectra of conductivity were obtained by the dielectric spectroscopy method. The authors analyzed the frequency dependences of the conductivity and observed a transition from the classical correlated barrier hopping (СВН) to quantum mechanical tunneling (QMT) at a certain temperature. Two areas with different types of conductivity were revealed in the conduction spectra, and the activation energies of charge transfer processes for amorphous and crystalline films were determined. The obtained results can be used to identify the amorphous phase in the quasi-binary chalcogenide system Sb2Te3-GeTe.
These results are original and important for the interpretation of the experimental data, leading to connection between the measured conductive properties and the structure of the material, however, I have some comments:
1. Introduction: please provide more recent citations.
2. line 48 has to be carefully checked, because instead of frequency “omega” in brackets a different symbol appears. Also, “at a particular frequency sigma0 “ means not frequency but conductivity:
“The total conductivity σtot(omega) measured in a given experiment at particular frequency σ0 and temperature can be written as:”
3. line 56: “where A is a constant dependent on temperature”…. In this case a constant has to be A(T), also in the formula...
4. line 51 “ac”, but line 58 “AC”: “Many different theories have been proposed for AC conduction in amorphous and crystalline semiconductors”…Please use the same lowercase or uppercase in the manuscript.
5. Barrier height WM appear in the line 73, while it is mentioned earlier in 69-71 lines: The relationship between the jump length Rω and the height of the potential barrier is expressed….Please indicate W meaning after or before the formula.
6. please correct the frequency symbol in 81 line: “R - jump length at a specific frequency ”, also 89, etc….
7. line 104: it is strange to see voltage written as “U = 10^(-1) V.” Why not 0.1 V?
8. y-axis in Figs.3,4: s has to be small, not capital.
9. Table 1: R(omega) depends on frequency (Eq(8), so it is not clear which frequency was used to determine values in the Table?
10. line 157: “The temperature dependence of conductivity in the coordinates σ(?) = ζ (103/T) for two samples is shown on figures 5 and 6”, but figures represent sigma’ .
11. line 176: please change the letter between parameters to English “and”
12. line 189 : you are analyzing films with thickness of 80 nm, but giving the result for 50 nm. Please explain in more details.
In general it is OK, but many small mistakes. For example: "The fallowing features" in the abstract.
Reviewer 2 Report
The paper is devoted for Sb2Te3 electrical conductivity investigations. The topic is generally interesting, however the paper contain unexplained places (below) and need major revisions.
At line 103 was indicated - measurements were performed in the frequency range 10-2 - 105 Hz, however the data in Figs.1-2 was presented only in frequency range 104-105 Hz why?
What behavior of DC conductivity of your samples?
Please explain the physical sense of conductivity activation energy at different frequencies (Fig.8)?
At which temperature is observed the semiconductor-metal transition in your samples (Fig.7)?
Why for investigation You have selected films with 80 nm thickness (line 94)? Why for investigations was selected voltage 10-1 V (line 104)?
Conclusions should be rewritten in more informative way. The last conclusions (lines 241-244) are not supported by discussion in the paper text.
Reviewer 3 Report
Authors have mainly presented the study of charge transfer processes in thin amorphous and crystalline Sb2Te3 films over the wide range of frequencies and temperatures. The work could be accepted after minor revision.
1. Introduction needs to be improved in relation to the problem of quasi-binary chalcogenide system Sb2Te3-GeTe, which is highlited in conlusion. Although there has been reports on this type of systems. Authors have reported the absence of interdiffusion between the crystalline GeTe and amorphous Sb2Te3 alloys, suggesting that their structural phase can change independently, based on the alloy composition. This type of quasi structure can be formed by two-phase change materials having different crystallization temperatures and reversible switching speed [Nanomaterials 2022, 12(10), 1623; https://doi.org/10.3390/nano12101623].
2. Figure 9b, should be reproduced with better resolusion.
Minor recheck required.
Round 2
Reviewer 2 Report
Authors make proper corrections according
to reviewer remarks and I suggest to publish the paper as it is.